# Associations between Circulating SELENOP Level and Disorders of Glucose and Lipid Metabolism: A Meta-Analysis

**DOI:** 10.3390/antiox11071263

**Published:** 2022-06-27

**Authors:** Ruirui Yu, Zhoutian Wang, Miaomiao Ma, Ping Xu, Longjian Liu, Alexey A. Tinkov, Xin Gen Lei, Ji-Chang Zhou

**Affiliations:** 1School of Public Health (Shenzhen), Shenzhen Campus of Sun Yat-sen University, Shenzhen 518107, China; yurr3@mail2.sysu.edu.cn (R.Y.); wangzht28@mail2.sysu.edu.cn (Z.W.); mamm8@mail2.sysu.edu.cn (M.M.); 2Shenzhen Health Development Research and Data Management Center, Shenzhen 518028, China; xuping8@mail2.sysu.edu.cn; 3Department of Epidemiology and Biostatistics, Dornsife School of Public Health, Drexel University, Philadelphia, PA 19104, USA; ll85@drexel.edu; 4Laboratory of Molecular Dietetics, IM Sechenov First Moscow State Medical University, 119146 Moscow, Russia; tinkov.a.a@gmail.com; 5Laboratory of Ecobiomonitoring and Quality Control, Yaroslavl State University, 150003 Yaroslavl, Russia; 6Department of Animal Science, Cornell University, Ithaca, NY 14853, USA; xl20@cornell.edu; 7Guangdong Province Engineering Laboratory for Nutrition Translation, Guangzhou 510080, China

**Keywords:** selenoprotein P, diabetes, glucose, lipid, low-density lipoprotein cholesterol, metabolic disorders, metabolic syndrome, non-alcoholic fatty liver disease, obesity

## Abstract

Selenoprotein P (SELENOP) is an extracellular antioxidant, selenium transporter, and hepatokine interfering with glucose and lipid metabolism. To study the association between the circulating SELENOP concentration and glucose and lipid metabolic diseases (GLMDs), including gestational diabetes (GD), metabolic syndrome (MetS), non-alcoholic fatty liver disease, obesity, and type 2 diabetes, as well as the individual markers, a meta-analysis was conducted by searching multiple databases from their establishment through March 2022 and including 27 articles published between October 2010 and May 2021, involving 4033 participants. Participants with GLMDs had higher levels of SELENOP than those without GLMDs (standardized mean difference = 0.84, 95% CI: 0.16 to 1.51), and the SELENOP levels were positively correlated with the markers of GLMDs (pooled effect size = 0.09, 95% CI: 0.02 to 0.15). Subgroup analyses showed that the SELENOP concentrations were higher in women with GD and lower in individuals with MetS than their counterparts, respectively. Moreover, SELENOP was positively correlated with low-density lipoprotein cholesterol, but not with the other markers of GLMDs. Thus, the heterogenicity derived from diseases or disease markers should be carefully considered while interpreting the overall positive association between SELENOP and GLMDs. Studies with a larger sample size and advanced design are warranted to confirm these findings.

## 1. Introduction

Globally, the prevalence of glucose and lipid metabolic diseases (GLMDs) is higher than that of all other diseases, posing serious threats to human health. The most frequently occurring GLMDs include dyslipidemia, gestational diabetes (GD), metabolic syndrome (MetS), non-alcoholic fatty liver disease (NAFLD) [1], obesity, and type 2 diabetes (T2D). They often coexist and may share a common pathophysiology [2]. Recent studies have shown that the liver, as the “hub” organ of GLMDs [3], can release several secreted proteins called hepatokines to regulate glucose and lipid metabolism (GLM) under stress conditions [4].

Selenoprotein P (SELENOP) is a liver-derived selenium (Se) carrier and a hepatokine in circulation. A single human SELENOP molecule contains up to ten selenocysteines covalently bounding Se. The first selenocysteine near the N-terminal of SELENOP has been demonstrated to have antioxidant properties, while the remaining nine concentrated at its C-terminal are mainly used to transport Se [5,6,7,8,9,10]. Studies have reported the relationship between SELENOP and several components and key indicators of GLMDs [11,12,13,14], but the results were inconsistent [4,15,16,17,18,19,20,21,22,23,24,25,26]. Although several meta-analyses have investigated the association of Se exposure [27,28] or Se supplementation [29,30] with the risk of GLMDs, none of the previous meta-analyses comprehensively examined the details of the association of SELENOP with major GLMDs, their individual components, and key indicators. In this study, we aimed to extend the previous studies by performing a further detailed meta-analysis for the purpose of adding to new evidence for future interventions with drugs and lifestyle factor changes to improve the prevention and control of the diseases.

## 2. Materials and Methods

### 2.1. Literature Search

This study was registered with PROSPERO in October 2021 and accepted for inclusion in November 2021 (registration ID number CRD42021257310), and the Preferred Reporting Items for Systematic reviews and Meta-Analyses (PRISMA) 2020 assertion [31] were strictly followed.

Two qualified investigators independently searched the following databases: PubMed, Embase, Web of Science, The Cochrane Library, China Biology Medicine, WanFang Data, VIP Database, and China National Knowledge Infrastructure, from their establishments to 14 March 2022. The searched keywords were (“selenoprotein P” OR “Selp” OR “Sepp” OR “Sepp1” OR “SELENOP”) AND (“glucose and lipid metabolism” OR “glucolipid metabolic disease” OR “dyslipidemia” OR “fatty liver” OR “obesity” OR “diabetes” OR “atherosclerosis” OR “hypertriglyceridemia” OR “metabolic syndrome” OR “glucose” OR “lipid” OR “insulin resistance” OR “fatty acid” OR “cholesterol” OR “triglycerides” OR “triglyceride” OR “body mass index” OR “BMI”), without restriction to any part of the publications. Furthermore, the references cited within the relevant articles were also reviewed in order to identify additional studies. All of the retrieved articles were published in English or Chinese and managed using the reference manager software EndNote X9 (Clarivate Analytics, Philadelphia, PA, USA).

### 2.2. Study Identification and Selection

Two independent reviewers performed the study identification and selection. Articles were included if they met all of the following criteria: (1) studies on GLMDs and GLM; and (2) presenting the concentrations of SELENOP in both patients and controls or at least one correlation coefficient between GLM indicators and SELENOP. The exclusion criteria were: (1) duplicate study on the same population; (2) irrelevant study judged from the title and abstract; (3) reviews, case reports, letters, editorials, abstracts, comments, and unpublished articles; (4) study with only animal or cellular experiment(s); (5) lack of predefined outcome data required for analyses; or (6) the evaluation score of the methodological quality (detailed in the next paragraph) being < 6. If there was a disagreement between the two independent researchers (R.Y. and Z.W.) for one study, its eligibility was reevaluated by a third investigator. The corresponding authors of studies with missing data or inaccessible full text were contacted.

### 2.3. Quality Assessment and Data Extraction

The Newcastle-Ottawa Scale [32,33] was used to assess the methodological quality of the finally included studies, with scores of 7–10 being high quality, 4–6 being moderate quality, and 0–3 being poor quality [34]. Following the inclusion and exclusion criteria, two of the authors assessed and extracted data from the studies independently after reading the key information of the articles, which included: (1) study characteristics, including author(s), year of publication, study country, ethnicity, disease, study design, adjusted factors, and sample size; (2) specimen, concentration, and detection method for SELENOP; and (3) type of correlation (Pearson correlation coefficient (PCC), Spearman correlation coefficient (SCC), or *r*^2^) between SELENOP and the GLM indicator. If an article had stratification analyses and reported SELENOP concentrations for different genotypes of a single nucleotide polymorphism (SNP), the reported concentrations were considered as those for different studies in our meta-analysis. However, if multiple correlation coefficient values were reported based on the same relationship between a GLM indicator and SELENOP, the total population or adjusted correlation coefficients were selected for inclusion. Disagreements in the data extraction process were resolved by group discussions between the authors.

### 2.4. Meta-Analysis

For the SELENOP concentration, data were expressed as the means and standard deviations (SDs); otherwise, the reported standard error (SE), median and interquartile range, or geometric mean and SE were converted to the expression by the corresponding formula [35,36,37,38]. For the correlation between SELENOP and GLM indicators, data were expressed as PCC, and SCC was converted to PCC using an appropriate method [39].

Standardized mean differences (SMDs) and 95% confidence intervals (CIs) were calculated to assess the differences in the SELENOP concentrations between groups. The associations between SELENOP and GLM indicators were evaluated through the combined correlation coefficients (*r*) (presented as the effect sizes (ESs) in the forest plots) and 95% CIs. The heterogeneity between studies was tested by calculating the Q statistic and the inconsistency index (*I*^2^) [40,41]. *p* < 0.05 or *I*^2^ > 50% indicated the presence of heterogeneity [42], and the random effect model was adopted. Otherwise, the fixed-effect model was used. Subgroup analysis was performed to examine possible sources of heterogeneity, and sensitivity analysis was conducted to detect the potential outliers. Publication bias was examined using funnel plots, Egger’s test, and Begg’s test.

Statistical analysis was performed using the STATA 15.0 software package (Stata Corporation, College Station, TX, USA). A two-sided *p* value of < 0.05 was considered statistically significant.

## 3. Results

### 3.1. Study Selection

In the first step of searching for published studies, 1967 articles were selected from the proposed search databases, and 1 [43] from the listed references in 1 of those articles [44]. After careful review, we excluded 607 duplicate articles and 1220 articles that did not meet the inclusion criteria (assessed by their study titles and abstracts). Of the 141 articles, we further excluded 114 articles without detail or eligible data for meta-analysis due to unsearchable full-texts (*n* = 11), animal or cellular studies (*n* = 13), reviews (*n* = 48), and others (*n* = 42). Finally, 27 articles [15,17,18,19,22,23,24,25,26,43,44,45,46,47,48,49,50,51,52,53,54,55,56,57,58,59,60] were included in the meta-analysis (Figure 1).

### 3.2. Study Characteristics

The selected articles for review were published between October 2010 and May 2021, covering 36 studies among different populations or those with different genotypes. The general characteristics of these studies are summarized in Table 1, with information on the relationships between the SELENOP level and the concerned indicators included in Appendix A. Of the studies, SMDs were extracted from 31 studies in the 23 articles, and correlation coefficients were obtained from 14 studies in the 13 articles (Appendix A). All of the included studies were of observational design, including 15 case-control studies in 9 articles and 21 cross-sectional studies in 18 articles. The sample sizes across the studies ranged from 21 to 905. The predominant method used to measure the concentrations of SELENOP was the enzyme-linked immunosorbent assay (ELISA). The high-performance liquid chromatography combined with inductively coupled plasma-mass spectrometry (HPLC + ICP-MS) method was adopted in two articles studying T2D, and the sol particle homogeneous immunoassay (SPIA) method was used in two articles studying hyperglycemia and overweightness/obesity, respectively. The SELENOP concentration data among the studies were presented as different statistics, though most were means ± SDs. For eight studies [17,18,19,24,45,52,53,59], the PCCs were calculated based on the SCCs provided in the papers (Appendix A).

Using the modified Newcastle-Ottawa scale [32,33], Appendix A shows the evaluation of the methodological quality of the selected 27 articles. Of them, 26 studies were considered as “high quality”, and the other one as “medium quality”. 

### 3.3. Meta-Analysis

A total of 4033 participants were involved in the included 27 articles, but in the meta-analysis, the number of participants (time per person) was 4292. In one article, participants were studied for different SNP genotypes [50], and another article provided data before and after follow-up [45].

#### 3.3.1. Relationships between SELENOP Level and GLMDs

The overall data revealed that the participants with GLMDs had significantly higher levels of SELENOP than the controls (SMD = 0.84, 95% CI: 0.16 to 1.51, *n* = 31). The heterogeneity test showed high heterogeneity among studies (*p*  <  0.001, *I*^2^ = 98.2%; Figure 2). However, the results of the sensitivity analysis indicated that, when each study was removed at a time, the overall random effect did not change significantly and no potential outliers were detected (Appendix A). Asymmetry was observed in the funnel plot (Appendix A), and statistical asymmetry tests also indicated the existence of publication bias (*p* = 0.003 for Egger’s test and 0.021 for Begg’s test).

Subgroup analyses based on the types of diseases showed that the level of SELENOP in the patients with GD was higher than that of the healthy controls (SMD = 9.47, 95% CI: 0.90 to 18.04, *I*^2^ = 99.2%, *n* = 3). The levels of SELENOP in the MetS patients were lower than those in the controls (SMD = −0.48, 95% CI: −0.65 to −0.30, *I*^2^ = 5.6%, *n* = 8), while the levels of SELENOP in the patients with other diseases were not different from those in the controls, respectively (Figure 2). However, if the sensitive study on NAFLD conducted by Zhang and Hao [54] (Appendix A) was excluded, the combined results showed a negative association between SELENOP and NAFLD (SMD = −0.97, 95% CI: −1.51 to −0.42, *I*^2^ = 84.6%, Appendix A).

In addition, subgroup analyses based on SELENOP detection methods were performed. The overall results showed that the SELENOP levels detected by the ELISA method in the GLMD patients were higher than those in the controls (SMD = 0.84, 95% CI: 0.13 to 1.55, *n* = 28) (Appendix A). Specifically, for T2D, both ELISA (SMD = 0.27, 95% CI: −1.82 to 2.35, *n* = 5) and HPLC + ICP-MS (SMD = −1.42, 95% CI: −3.07 to 0.24, *n* = 2) did not find any significant differences in the SELENOP levels between the patients and controls (Appendix A). For obesity, the ELISA found no significance (SMD = 0.38, 95% CI: −0.02 to 0.77, *n* = 5), but the SPIA indicated higher SELENOP levels in patients than in the controls (SMD = 22.96, 95% CI: 15.80 to 30.13, *n* = 1) (Appendix A).

#### 3.3.2. Correlations between SELENOP and GLM Markers

Figure 3 indicates that SELENOP was positively and significantly correlated with the body mass index (BMI), fasting insulin (FIns), fasting plasma/serum glucose (FPG), hemoglobin A1c (HbA1c), high-density lipoprotein cholesterol (HDL-C), homeostasis model assessment of insulin resistance (HOMA-IR), low-density lipoprotein cholesterol (LDL-C), total cholesterol (TC) [22], and triglyceride (TG) (pooled ES = 0.09, 95% CI: 0.02 to 0.15, *n* = 75). The inter-study heterogeneity across the studies was significant (*I*^2^ = 92.1%, *p* = 0.000, Figure 3). In the subgroup analysis, a positive and significant correlation was only observed between SELENOP and LDL-C (pooled ES = 0.14, 95% CI: 0.01 to 0.27, *n* = 5, *I*^2^ = 73.7%, *p* = 0.004), but there were significant correlations observed with any other GLM markers. Sensitivity analyses were performed, and the results remained consistent with the pooled effect.

## 4. Discussion

The plasma/serum Se includes that from SELENOP, glutathione peroxidase (GPX) 3, albumin-bound fraction, free Se, etc. Containing multiple selenocysteine residues, SELENOP accounts for 48–53% of the blood Se [10,61]. Se deficiency may lead to a decrease in SELENOP expression or premature terminations just before one of the selenocysteine residues in the C-terminal, resulting in SELENOP truncates [9,10]. On the other hand, the SELENOP expression may become saturated at a certain high Se level and resist a further increase in the Se level [9]. Therefore, the SELENOP protein abundance assayed using the predominant methods targeting the N-terminal did not parallel the circulation Se concentration, and SELENOP was not representable by Se in terms of the circulating concentration related to GLMDs [45]. Unfortunately, there were few studies meeting our inclusion criteria presenting both Se and SELENOP data, which prevented us from comparing their differences in association with the outcomes by reliable quantification methods, instead of general discussions. More interestingly, SELENOP is cleavable with two kallikrein cutting sites between the first and the second selenocysteine residues [10]. It has been reported that proteases are activated at inflammatory sites [62], and that Se modulates the inflammatory and immune responses [63]. Therefore, it is possible that plasma-kallikrein-processed SELENOP fragments regulate inflammation by controlling Se action in cells, though the cleaved form is about 1.2% of the total SELENOP in the human plasma [64].

Studies have shown that a decrease in SELENOP may cause various dysfunctions related to Se deficiency and oxidative stress [65,66,67,68]. Furthermore, low SELENOP concentrations were strongly associated with the risk of incident cardiovascular disease and mortality from all causes, cardiovascular disease, and COVID-19 [12,69]. Nevertheless, excessive SELENOP may lead to insulin resistance [70], and treatment with the full-length form of SELENOP may impair insulin signal transduction in cultured hepatocytes [46]. In animal experiments, an excess of circulating SELENOP induces both impaired insulin signaling in the peripheral tissues and decreased insulin secretion in the pancreas [71]. Moreover, SELENOP may induce insulin resistance by affecting adipose tissue to reduce the adiponectin levels [72] or by acting on cultured myotubes with low-density lipoprotein (LDL) receptor-associated protein 1 [73], thereby leading to diabetes. On the other hand, insulin exerts inhibitory effects on the gene expression of SELENOP in hepatocytes [46,74,75], and impaired insulin action in certain metabolic disorders might increase the expression and circulating level of SELENOP. Therefore, the overproduction of SELENOP and the development of metabolic disorders might reinforce one another mutually in a vicious cycle, and SELENOP is a multifunctional protein in the pathology of GLMDs [9,76].

### 4.1. GD and T2D

During a normal pregnancy, an increase in insulin resistance occurs simultaneously with an increase in oxidative stress, which is particularly prominent in women with GD [77]. Growing attention has been paid to the association between SELENOP and GD [78,79,80,81,82,83,84]. Two recent meta-analyses on Se and GD observed that the serum Se levels in GD patients were significantly lower than those in the healthy pregnancy group [85,86]. However, the results of our meta-analysis showed that the SELENOP concentrations were elevated in patients with GD compared with normal pregnant women, which is consistent with a recently published meta-analysis on the relationship between hepatokines (including only one article for SELENOP) and GD [87]. Though one of the three included articles in our meta-analysis tested both the circulating Se and SELENOP concentrations [15], it seems that there were no obvious clues to understand the relationship between Se and SELENOP in their associations with GD by indicating no difference in the Se levels (76–78 µg/L), but SELENOP in GD women was lower compared with their counterparts.

One study showed that the SELENOP concentration at baseline in the oral glucose test was associated with the future post-load plasma glucose in male participants, whereas, in female participants, it was interrelated with the future FPG [45]. This suggests that sexual dimorphism is present in the main target organs of SELENOP in humans, and it may help to interpret the positive correlation of SELENOP with GD characterized by upregulated hepatic glucose production.

The overall results of the nine T2D studies included in our meta-analysis [18,19,43,44,56] showed that the SELENOP concentrations in T2D were similar to those without T2D, which was inconsistent with the results of the previous meta-analysis studies on the relationship between Se and T2D [27,28,88,89], suggesting disparity between Se and SELENOP in their associations with diseases. In previous studies, the investigators suggested that there was a U-shaped relationship between Se and T2D. Either a significantly decreased or elevated Se level, lower or higher than the normal physiological range, should be considered as a risk factor for T2D [90,91,92,93]. Of the two included studies for our meta-analysis that simultaneously measured the concentrations of Se and SELENOP [43,44], both Se and SELENOP did not differ between the patients and controls when the mean plasma Se was about 80 µg/L [43], but both were significantly higher in patients than in the controls when the mean serum Se was > 94 µg/L [44]. However, in another recently published cohort study that did not meet our inclusion criteria, the serum Se (with median value at 80 µg/L), but not SELENOP, was associated with increased risk of developing T2D in the final adjustment model [94]. Thus, the circulating Se levels may affect the correlation between SELENOP and T2D, but this needs to be further verified by more studies.

### 4.2. MetS and NAFLD

MetS is a complex disease defined by a set of interrelated metabolic factors [95]. The combined results of the articles included in this study showed that lower SELENOP levels were associated with increased risk of MetS, which is consistent with the findings of a recently published review [96]. Though the only article included in our meta-analysis on MetS presenting both Se and SELENOP data suggested no difference in the serum Se and lower SELENOP in patients than in controls [50], another meta-analysis on the association between the dietary Se level and MetS indicated their negative association [97].

Two articles that were not included in our study due to data skewness also showed lower SELENOP concentrations in MetS patients [52,98]. Patients with MetS are usually in an inflammation state, coupled with impaired liver functions [99]. Thus, the circulating SELENOP, as a negative acute-phase reactant and a hepatokine, could be downregulated by both inflammation [51,100,101,102] and decreased selenoprotein synthetic capacity in the MetS status.

NAFLD is considered to be the hepatic symptom of MetS due to its coexistence with visceral obesity, insulin resistance, and dyslipidemia [1], and its primary characteristic is the accumulation of lipids in the liver accompanied by lipid peroxidation, oxidative stress, inflammation, etc. [103]. Though SELENOP has been shown to play an important role in the pathological process of lipid accumulation [104], the relationship between SELENOP and NAFLD remains unclear. Our sensitivity analysis on the NAFLD subgroup of GLMDs indicated that the study conducted by Zhang and Hao [54] may affect the stability of the null association between SELELOP and NAFLD, though it did meet our inclusion criteria. If the study was excluded, the lower SELENOP levels in NAFLD patients were consistent with the relationship between MetS and SELENOP. However, the SELENOP concentrations were higher in pregnant women with NAFLD and increased the risk of GD [16], which suggested that some specific physiological conditions may complicate the association. Other confounding variables may also add to the heterogeneity between studies and affect their results. Furthermore, the higher SELENOP levels in NAFLD in some studies may imply a protective mechanism that counteracts the higher oxidative stress in the initial stage of diseases, but the mechanism may be insufficient in the advanced stages, such as definite non-alcoholic steatohepatitis [24], cirrhosis [105], and hepatocellular carcinoma [106], in which lower SELENOP levels have been reported.

### 4.3. Obesity and BMI

Oxidative stress in obesity leads to metabolic and endocrine dysfunction of adipose tissue, contributing to the development of obesity-related insulin resistance [107,108]. SELENOP has been shown to respond significantly to this proinflammatory stimulus [96]. A previous review pointed out that the circulating SELENOP levels were elevated in patients with obesity [96], but our meta-analysis of the included studies found no difference in the SELENOP levels between patients with obesity and healthy subjects, further confirming a recent meta-analysis on the Se (*n* = 14 for children and 18 for adults) and SELENOP (*n* = 3, which were among our included 6 studies) levels in people with overweightness and obesity [109]. The most likely reason is that the individuals with obesity were not able to be separated from the overweight and obese individuals, because almost all of the included studies defined the cases with BMI > 25 kg/m^2^ for adults. If subjects with obesity defined by a higher cut-off value of BMI were compared with normal-weight subjects, a statistically significant relationship between obesity and the SELENOP levels might be observed. In addition, two of the included studies recruited individuals with anemia [47] or NAFLD [25], which might also have had a certain impact on the results.

BMI is commonly used to define overweightness and obesity in clinical settings and in epidemiological studies [110]. As a measure of relative weight, it is directly linked to health risks and mortality in many populations [111]. Both BMI [112] and SELENOP are associated with insulin resistance and inflammation, and the relationship between BMI and SELENOP is still controversial. The results of our meta-analysis showed that SELENOP was not statistically correlated with BMI, but the heterogeneity was high. The possible reasons are as follows: the included subjects were not all obese, and the distributions of age, race, and gender of each study population varied, which might affect the levels of SELENOP and BMI, as well as their relationship [45,113].

### 4.4. Lipid Profiles

Meta-analyses have shown that Se supplementation does not affect the lipid levels or only results in a statistically significant improvement in the TC, TG, and/or VLDL-C levels [29,30]. A similar kind of inconsistency also existed in the association between SELENOP and the lipid profiles. Some evidence showed that SELENOP was positively correlated with HDL-C [17,21,48,52], LDL-C [26,58], TC [22,52], and TG [18,21,55] in the plasma/serum. In contrast, some cross-sectional studies reported that higher SELENOP levels were correlated with lower HDL-C [55], TC [48,58], and TG [48,52,58], while others indicated no correlation with LDL-C [18,52]. Given the paradox, we conducted a comprehensive meta-analysis of the relationship between SELENOP and lipid profiles. The results showed that SELENOP was positively correlated with LDL-C, but not with other lipid indices. 

LDL, carrying cholesterol in the form of LDL-C, circulates in the plasma and supplies various cells with cholesterol under normal physiological conditions, but oxidized LDL has cytotoxic effects and is thought to be involved in the development of atherosclerosis, a type of GLMD [114]. Having the ability to reduce phospholipid hydroperoxides and bind to glycosaminoglycans [115], SELENOP may play an antioxidant protective role by binding to ApoB-100 [116], a glycosylated LDL component, and SELENOP can protect LDL against oxidation in a cell-free in vitro system [114]. In response to the oxidized LDL potentially existing in diseases with high LDL-C levels included in our study, SELENOP might be elevated. In addition, studies have suggested that the role of SELENOP in GD and T2D might reduce the adiponectin levels by affecting adipose tissue [20,72], and the adiponectin levels are known to be negatively correlated with LDL-C [117]. This can also account for the positive relationship between SELENOP and LDL-C.

### 4.5. Glucose Metabolism

Se and SELENOP are closely related to glucose metabolism. For Se, a recent study showed a linear relationship between Se deficiency and hypoglycemia in healthy adults. The mechanism is suggested to be the low Se status potentially causing diminished activity of GPX1 (one of the selenoproteins) in insulin target cells, contributing to amplified insulin signals due to the dysregulation of redox-regulated proteins, such as the insulin-antagonistic protein tyrosine phosphatase 1B (PTP1B). In contrast, an elevated Se concentration upregulates PTP1B and induces suppressed insulin signaling and hyperglycemia [118]. A certain concentration of serum Se is essential to maintain the expression of GPX or SELENOP, which seems to be required for achieving and maintaining euglycemia. The relationship between SELENOP and serum glucose in the Se-deficient state needs further study. Furthermore, a high glucose concentration stimulates the pancreas to secrete insulin and the liver to release SELENOP [53], while excessive SELENOP worsens glucose metabolism via insulin resistance and the impairment of insulin secretion [18,46,71]. Therefore, the causal relationship between an increased SELENOP concentration and metabolic disorders of glucose has not been clarified [119]. Moreover, the correlations between SELENOP and glucose metabolism indicators (FIns, FPG, HbA1c, and HOMA-IR) have not been uniformly established yet. 

Elevations in FPG and HbA1c reflected acute dysregulated glucose metabolism and chronic hyperglycemia, respectively [120], and HOMA-IR, another glucose metabolism indicator, is based on insulin resistance and hyperinsulinemia [121]. A previous cross-sectional study showed that moderate positive correlations were observed between the SELENOP levels and FPG and HbA1c among T2D patients [20], and the hepatic mRNA abundance of *SELENOP* was also positively correlated with FPG [46]. In addition, a Korean study reported that serum SELENOP was positively correlated with HOMA-IR in NAFLD patients [21]. In contrast, more recent epidemiologic findings showed inverse relationships between the SELENOP levels and FIns, FPG, HbA1c, HOMA-IR, and several other metabolic traits in adults [49,52] and young children [48]. However, a follow-up study found no significant associations between the baseline serum SELENOP concentrations and these metabolic markers [45]. We pooled these inconsistent results and found that SELENOP was not associated with indicators of glucose metabolism in the population, which suggested that the concentrations of SELENOP, FPG, and HbA1c for participants of a healthy condition and with various diseases, as well as their different stages, may not achieve a consistent correlation.

### 4.6. SNPs of SELENOP

The SNPs of the *SELENOP* gene were found to be correlated with certain metabolic phenotypes. In a meta-analysis that included three different ethnic groups, rs28919926 and rs146125471 showed associations with acute insulin resistance, and rs7579 with the insulin sensitivity index [122]. The rs7579 A allele is associated with a decrease in the SELENOP levels in subjects with or without MetS, and MetS decreases the SELENOP levels in general, except for the rs7579 AA homozygote carriers. However, rs3877899 AA coordinates with MetS to decrease the SELENOP levels [50]. In Turkish pregnant women, the rs13154178 GG genotype was coupled with a higher SELENOP concentration in GD patients [15], and the G allele was positively associated with FPG and GD occurrence [72]. Another study in pregnant women from the United Kingdom showed that, under Se supplementation during pregnancy, the rs3877899 A allele helped to maintain the Se status at a constant level, though the activity of GPX3, another circulating selenoprotein, increased [123]. Moreover, *SELENOP* rs3877899 altered the LDL levels in response to Brazil nut intake, suggesting that *SELENOP* polymorphisms affected the ability of Se to improve lipid biomarkers [124]. In general, very few studies have considered variations in SELENOP in its association with GLMDs, which needs more attention in the future.

### 4.7. Limitations and Advantages

Several limitations of our study should be kept in mind while interpreting the results. First, the studies used different laboratory methods to test the SELENOP concentrations. For example, results from ELISA, HPLC + ICP-MS, and SPIA may lead to high heterogeneity in the synthesis results. Second, potential publication bias may have occurred. Several reviewed studies in the report had a small sample size, and several studies applied a cross-sectional study design, which is unable to test any causality. Third, we could not obtain the original datasets of the reviewed studies, which made it impossible to use the ab initio analysis of data to conduct analysis in detail, such as to test a dose-response relationship. Fourth, unobserved bias may occur due to the included studies with different confounding factors adjusted.

Despite the limitations discussed above, this study has two important strengths. First, the study focusing on the relationships of SELENOP with several GLMDs and related indicators is novel. Thus, building upon the recently published studies, our meta-analysis study with a robust design provides updated evidence on SELENOP studies in the body of literature. Second, we used SMD and correlation-coefficient-derived ES to evaluate the relationships and examined multiple factors. These analysis approaches enhanced the statistical power and led to the results of our study being more explanatory.

## 5. Conclusions

In conclusion, among the major GLMDs, there was a positive correlation between increased circulating SELENOP concentration and the risk of GD and elevated LDL-C concentration, but a negative correlation with MetS. Further epidemiological studies with a larger sample size, advanced study designs, and especially comparison of Se and SELENOP in their associations with GLMDs are needed to test the specific causal association between SELENOP and GLMDs and/or SELENOP’s value for the prediction and treatment of GLMDs.

## Figures and Tables

**Figure 1 antioxidants-11-01263-f001:**
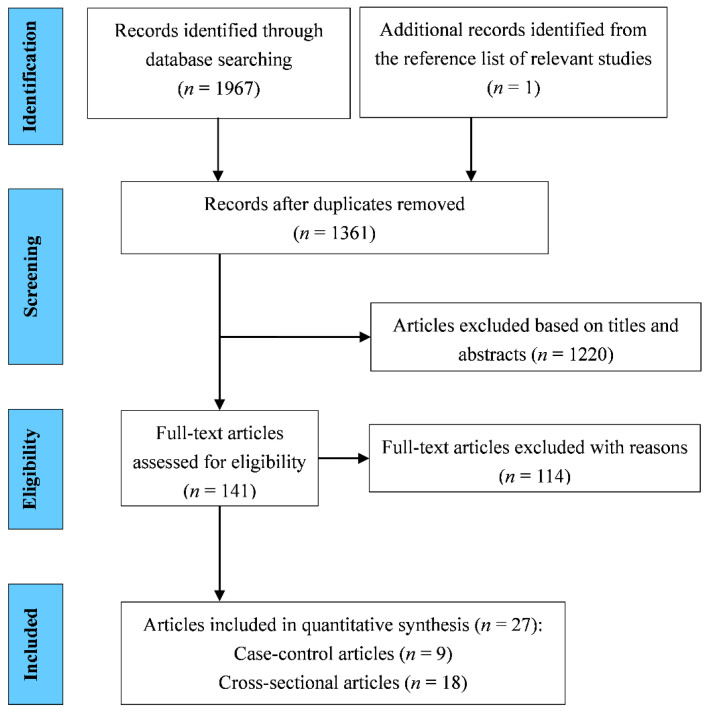
Strategy for literature search and selection.

**Figure 2 antioxidants-11-01263-f002:**
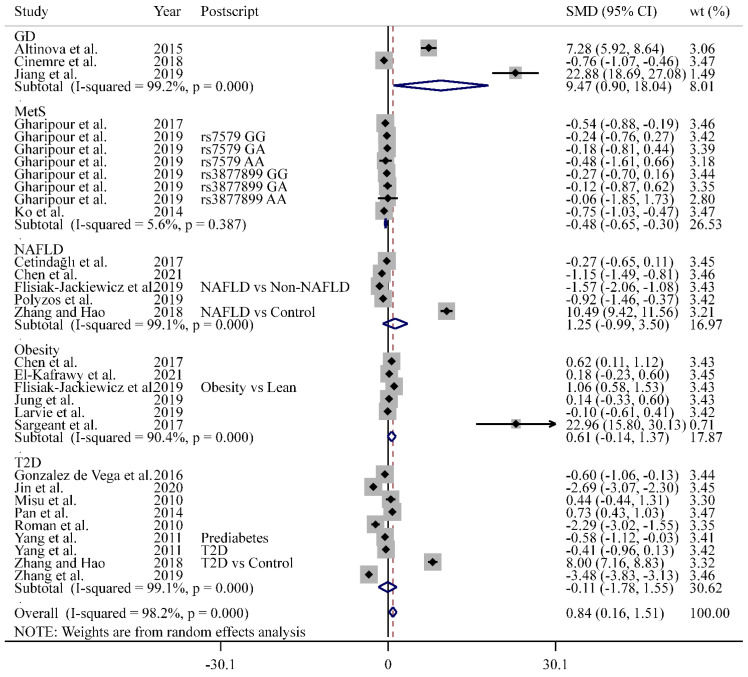
Correlations between the circulating selenoprotein P level and disorders of glucose and lipid metabolism. Abbreviations: GD, gestational diabetes; MetS, metabolic syndrome; NAFLD, non-alcoholic fatty liver disease; SMD, standardized mean difference; T2D, type 2 diabetes; wt, weight.

**Figure 3 antioxidants-11-01263-f003:**
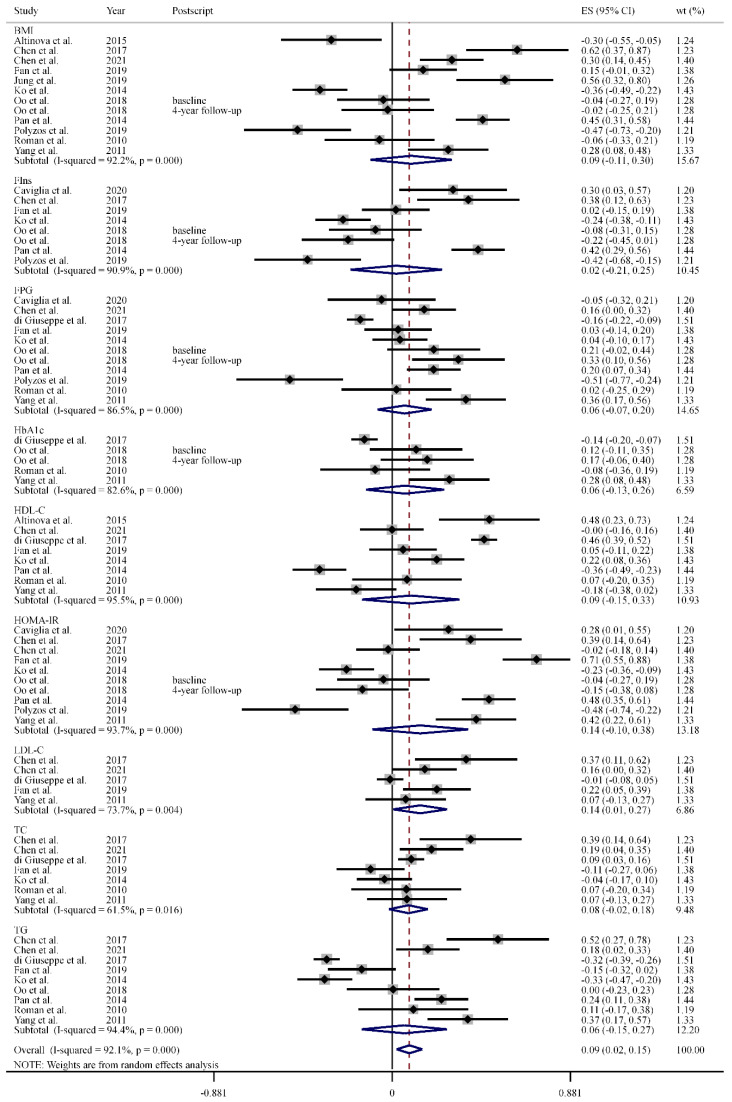
Correlations between selenoprotein P and 9 GLM markers. Abbreviations: ES, effect size; FIns, fasting insulin; FPG, fasting plasma/serum glucose; GLM, glucose and lipid metabolism; HDL-C, high-density lipoprotein cholesterol; LDL-C, low-density lipoprotein cholesterol; TC, total cholesterol; TG, triglyceride; wt, weight.

**Table 1 antioxidants-11-01263-t001:** Characteristics of the selected studies for meta-analysis.

Study *^a^*	Country	Disease	*n*		Sample (Unit)	Detection Method *^b^*	Level *^c^*	
			Case	Control			Case	Control
* Altinova et al., 2015 [17]	Turkey	GD	30	35	Plasma (ng/mL)	ELISA ^2^	6.2 (4.5–8.2) ^♦^	7.9 (4.5–10.7) ^♦^
* Caviglia et al., 2020 [53]	Italy	NAFLD	57		Serum (ng/mL)	ELISA ^1^	T3: 11.8	
# Cetindağlı et al., 2017 [26]	Turkey	NAFLD	93	37	Plasma (ng/mL)	ELISA ^9^	1574.2 ± 972.1 ^♠^	232.7 ± 371.05 ^♠^
* Chen et al., 2017 [22]	Australia	OW/OB	34	29	Plasma (μg/mL)	ELISA ^1^	52.3 ± 39.1 ^♠^	14.5 ± 12.8 ^♠^
# Chen et al., 2021 [59]	China	NAFLD	79	79	Serum (μg/mL)	ELISA ^1^	13.4 ± 7.0 ^♠^	11.1 ± 7.1 ^♠^
# Cinemre et al., 2018 [15]	Turkey	GD	86	90	Plasma (ng/mL)	ELISA ^8^	35.29 ± 3.00 ^♣^	46.98 ± 4.59 ^♣^
* di Giuseppe et al., 2017 [52]	Germany	MetS	Q1: 225; Q2: 227; Q3: 228; Q4: 225		Serum (mg/mL)	ELISA ^2^	Q1: 2.86 (1.96–3.70) ^♦^; Q2: 4.52 (3.87–5.98) ^♦^; Q3: 6.05 (5.3–28.47) ^♦^; Q4: 11.72 (8.07–15.79) ^♦^	
* El-Kafrawy et al., 2021 [51]	Egypt	OW/OB	50	40	Serum (mg/L)	ELISA ^7^	16.18 ± 3.99 ^♠^	4.25 ± 4.27 ^♠^
* Fan et al., 2019 [58]	China	T2D and NAFLD	T2D and NAFLD: 79; T2D: 61		Serum (ng/mL)	ELISA ^1^	T2D and NAFLD: 1341.11 ± 290.51 ^♠^; T2D: 755.77 ± 184.90 ^♠^	
* Flisiak-Jackiewicz et al., 2019 [25]	Poland	NAFLD Obesity	34 86	52 24	Serum (pg/mL)	ELISA ^1^	19449.5 (13327–28058) ^♦^ 21421 (11566–28058) ^♦^	21629 (10369.5–27976) ^♦^ 5411 (1618–15135) ^♦^
* Gharipour et al., 2017 [49]	Iran	MetS	65	71	Serum (ng/mL)	ELISA ^3^	41.8 ± 6.57 ^♣^	81.5 ± 15.2 ^♣^
# Gharipour et al., 2019 [50]	Iran	MetS	rs7579 GG: 29	30	Serum (ng/mL)	ELISA ^3^	55.52 ± 16.78 ^♣^	109.48 ± 29.78 ^♣^
			rs7579 GA:18	22			36.65 ± 7.41 ^♣^	59.80 ± 22.06 ^♣^
			rs7579 AA: 8	5			29.45 ± 1.97 ^♣^	26.65 ± 2.51 ^♣^
			rs3877899 GG: 40	44			40.37 ± 8.44 ^♣^	83.91 ± 21.33 ^♣^
			rs3877899 GA: 15	13			56.92 ± 23.34 ^♣^	86.42 ± 40.99 ^♣^
			rs3877899 AA: 2	3			29.70 ± 4.1 ^♣^	81.95 ± 107.03 ^♣^
# Gonzalez de Vega et al., 2016 [43]	Spain	T2D	78	24	Plasma (ppb)	HPLC + ICP-MS	41.9 ± 12.6 ^♠^	50.5 ± 19.1 ^♠^
# Jiang et al., 2019 [57]	China	GD	30	30	Serum (mmol/L)	ELISA ^1^	4.85 ± 1.02 ^♠^	2.43 ± 1.04 ^♠^
# Jin et al., 2020 [56]	China	DN	100	100	Serum (ng/mL)	ELISA ^1^	673.18 ± 86.94 ^♠^	973.84 ± 132.27 ^♠^
* Jung et al., 2019 [60]	Korea	OW/OB	35	35	Serum (μg/mL)	ELISA ^2^	2.3 ± 0.1 ^♣^	1.5 ± 0.1 ^♣^
* Ko et al., 2014 [48]	Korea	MetS	94	116	Serum (ng/mL)	ELISA ^2^	16.7 ± 2.2 ^♥^	28.6 ± 2.0 ^♥^
* Larvie et al., 2019 [47]	America	OW/OB	32	27	Plasma (ng/mL)	ELISA ^4^	352.13 (276, 446) ^♥^	360.77 (290, 450) ^♥^
* Misu et al., 2010 [46]	Japan	T2D	12	9	Serum (μg/mL)	ELISA ^9^	6.7 ± 0.9 ^♣^	5.1 ± 1.7 ^♣^
* Oo et al., 2018 [45]	Japan	HG	76		Serum (μg/mL)	SPIA	Baseline: 2.51 ± 0.52 ^♠^	
* Pan et al., 2014 [55]	China	T2D	156	64	Serum (mmol/L)	ELISA ^1^	3.77 ± 1.79 ^♠^	2.34 ± 2.30 ^♠^
* Polyzos et al., 2019 [24]	Greece	NAFLD	31	27	Serum (mg/L)	ELISA ^5^	SS: 4.2 ± 0.3 ^♣^; Borderline NASH: 4.1 ± 0.4 ^♣^; Definite NASH: 3.0 ± 0.5 ^♣^	5 ± 0.2 ^♣^
* Roman et al., 2010 [19]	Italy	T2D	40	15	Plasma (ng/mL)	HPLC + ICP-MS	58 ± 9 ^♠^	56 ± 8 ^♠^
* Sargeant et al., 2017 [23]	Britain	OW/OB	11	11	Plasma (μg/mL)	SPIA	2.81 ± 0.30 ^♠^	3.01 ± 0.39 ^♠^
* Yang et al., 2011 [18]	Korea	T2D PreD	40 40	20	Serum (ng/mL)	ELISA ^1^	1032.4 (495.9–2149.4) ^♦^; 867.3 (516.3–1582.7) ^♦^	62.0 (252.5–694.5) ^♦^
# Zhang and Hao, 2018 [54]	China	T2D NAFLD	100 100	100 100	Serum (mmol/L)	ELISA ^6^	3.05 ± 1.20 ^♠^ 4.42 ± 1.80 ^♠^	2.33 ± 2.30 ^♠^ 2.33 ± 2.30 ^♠^
# Zhang et al., 2019 [44]	China	T2D	176	142	Serum (ng/mL)	ELISA ^1^	1811.1 ± 36.3 ^♣^	1688.2 ± 40.5 ^♣^

Note: DN, diabetic nephropathy; ELISA, enzyme-linked immunosorbent assay; GD, gestational diabetes; HG, hyperglycemia; HPLC, high-performance liquid chromatography; ICP-MS, inductively coupled plasma-mass spectrometry; MetS, metabolic syndrome; NAFLD, non-alcoholic fatty liver disease; NASH, non-alcoholic steatohepatitis; OW/OB, overweight and obesity; PreD, prediabetes; SPIA, sol particle homogeneous immunoassay; SS: simple steatosis; T2D, type 2 diabetes; *n*, sample size number. *^a^* *, cross-sectional study; #, case-control study. *^b^* ELISA kits were provided by ^1^, Cloud-Clone Corp. Houston, TX, USA; ^2^, Cusabio, Wuhan, China; ^3^, Eastbiopharm, Hangzhou, China; ^4^, MyBioSource (San Diego, CA, USA); ^5^, selenOmed GmbH, Berlin, Germany; ^6^, Shanghai Runyu Biotechnology Co., Ltd., Shanghai, China; ^7^, Shanghai Sunred Biological Technology Co., Ltd., Shanghai, China; ^8^, Shanghai YeHua Biological Technology Co., Ltd. Gical Technology Co., Ltd., Shanghai, China; ^9^, unknown. *^c^* Data were expressed as quartiles (Q1/2/3/4), tertiles (T1/2/3), medians (interquartile ranges) (^♦^), means ± SDs (^♠^), means ± SEs (^♣^), or geometric means ± SDs (^♥^) for all subjects or patients vs controls.

## Data Availability

The data presented in this study are available in the article and Appendix A.

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
