# Peer review of "Associations between Circulating SELENOP Level and Disorders of Glucose and Lipid Metabolism: A Meta-Analysis"

_antioxidants, 2022, doi:10.3390/antiox11071263_

Round 1
Reviewer 1 Report
I accepted the manuscript of Ruirui Yu et al. for review having great expectations. A clear review of the associations between circulating SELENOP level and disorders of glucose and lipid metabolism would be valuable to the field.
However, I do not see the added value of the manuscript as presented. My biggest concern is the lack of biological explanation/proof. It would have been a great opportunity to assess studies in which both serum Se and Selenop were measured. Or to compare outcome between studies build on SELENOP on one side and serum Se on the other. For example: https://doi.org/10.1016/j.redox.2020.101709. In this studie the authors showed the linear association of Se deficiency with hypoglycemia in healthy adults (N=over 5000!! serum Se measurements). If the authors of the current manuscript would have used this to validate their results, that would have been of value.
Author Response
Thank the reviewer’s comments in the Report Form. We've tried to make some changes to our best.
Response to "Comments and Suggestions for Authors
":
Thank you for the important suggestion. For there has been few studies meeting our inclusion criteria to present both Se and SELENOP data that allow us to compare the outcomes based on Se and SELENOP, we only add the reviewer’s suggestion in Discussion section (Lines 233-235 and 353-359).
Reviewer 2 Report
This is an excellent paper with reliable analyses.
1. Does the SELENOP level predict the future mortality by all causes and GLMDs-related complications? More descriptions can be added to the current manuscript.
2. In Abstract (row/line 32), Before the last sentence (starting ‘studies with’), the suggestion from the results can be added.
3. The positive association of SELENOP with LDL-C might be due to the increase of antioxidant capacity under an increased oxidative stress burden with LDL. On the other hand, the negative association of SELENOP with MetS traits might be due to the decrease of antioxidant capacity under an increased oxidative stress burden with MetS. Some readers may think the interpretation as a convenience, double standards or opportunism. About it, more discussion may be added.
4. As the authors mentioned, we could find the inconsistent data across measurement methods (ELISA and other methods) even if the blood specimens were the same. The sub-analyses by methods can be performed and presented.
5. In Methods (row/line 87), who were the two independent researchers? Their initial names can be added.
Author Response
Thank the reviewer’s comments in the Report Form. We’ve tried to make some changes to our best.
Responses to "Comments and Suggestions for Authors
":
Point 1: Does the SELENOP level predict the future mortality by all causes and GLMDs-related complications? More descriptions can be added to the current manuscript.
Response 1: Thanks for the comments. We have added more descriptions in the revised version (Lines 243-245).
Point 2: In Abstract (row/line 32), Before the last sentence (starting ‘studies with’), the suggestion from the results can be added.
Response 2: The suggestion is added and some other words are deleted as limited by the 200 words for the abstract.
Point 3: The positive association of SELENOP with LDL-C might be due to the increase of antioxidant capacity under an increased oxidative stress burden with LDL. On the other hand, the negative association of SELENOP with MetS traits might be due to the decrease of antioxidant capacity under an increased oxidative stress burden with MetS. Some readers may think the interpretation as a convenience, double standards or opportunism. About it, more discussion may be added.
Response 3: Thank you for pointing out the apparent conflict interpretions, and we give more discussion. As mentioned in the introduction, the first selenocysteine near the N-terminal of SELENOP is demonstrated to have antioxidant property, and the elevated SELENOP level may indicate a protective mechanism against the higher oxidative stress. During this process, SELENOP is consumed and down-regulated as the disease progresses or inflammation increases, such as MetS. In addition, studies have shown that MetS can cause liver inflammation and damage to liver function, which in turn reduces the synthesis of SELENOP. This is different from the positive correlation between LDLC and SELENOP, because elevated LDLC does not necessarily result in liver function injury. We have added corresponding content in the discussion (Lines 287-290).
Point 4: As the authors mentioned, we could find the inconsistent data across measurement methods (ELISA and other methods) even if the blood specimens were the same. The sub-analyses by methods can be performed and presented.
Response 4: It is absolutely a good reminding that measurement methods may confuse the relationships between SELENOP level and GLMDs. We added some description for the methods in Lines 147-151 and subgroup analyses stratified by ELISA, HPLC+ICP-MS, and SPIA for GLMDs in Lines 193-201 and Supplementary Figure S6-S8. Higher SELENOP levels were found in GLMD groups by ELISA and SPIA subgroups but not in HPLC+ICP-MS subgroup (Supplementary Figure S6). When the specific diseases considered, only T2D and obesity used more methods than ELISA, and subgroup analyses only found that the SPIA method got a significance in only one study for obesity by Sargeant et al. (another study using SPIA by Oo et al. did not present the data for the case and control groups).
Point 5: In Methods (row/line 87), who were the two independent researchers? Their initial names can be added.
Response 5: Their initial names have been added on the updated Lines 88. Thanks!
Round 2
Reviewer 1 Report
No improvement was made, other than 3 sentences in the discussion. This has not improved the scientific value of the manuscript and I believe that it is therefore not suitable for publication
Author Response
Thank you very much for your comments, your suggestions are very forward looking. We’ve tried to make more changes to our best.
Please check the uploaded responses in detail.
Some other changes including language are made in the tracking model in the revised manuscript.

Reviewer 2 Report
The suggested parts were explained in the revision. In Abstract, the abbreviation ‘LDL-C’ needed?
Author Response
Thank you for your positive comments on our manuscript. We are very fortunate that you reviewed this manuscript.
Responses to "Comments and Suggestions for Authors
":
Point 1: The suggested parts were explained in the revision. In Abstract, the abbreviation ‘LDL-C’ needed?
Response 1: Thanks for reminding us. We have deleted the abbreviation of LDL-C (Line 31).